🔓 | Open Peer Review | Eukaryotic Cells | Research Article

# Involvement of 2-deoxyglucose-6-phosphate phosphatases in facilitating resilience against ionic and osmotic stress in *Saccharomyces cerevisiae*

Chinmayee Awasthy,[1] Zeinab Abdelmoghis Hefny,[1] Wouter Van Genechten,[1] Uwe Himmelreich,[2] Patrick Van Dijck[1]

**ABSTRACT** The *Saccharomyces cerevisiae DOG* genes, *DOG1* and *DOG2*, encode for 2-deoxyglucose-6-phosphate phosphatases. These enzymes of the haloacid dehalogenase superfamily are known to utilize the non-natural 2-deoxyglucose-6-phosphate as their substrate. However, their physiological substrate and hence their biological role remain elusive. In this study, we investigated their potential role as enzymes in biosynthesizing glycerol through an alternative pathway, which involves the dephosphorylation of dihydroxyacetone phosphate into dihydroxyacetone, as opposed to the classical pathway which utilizes glycerol 3-phosphate. Overexpression of *DOG1* or *DOG2* rescued the osmotic and ionic stress-sensitive phenotype of *gpp1Δ gpp2Δ* or *gpd1Δ gpd2Δ* mutants, both affected in the production of glycerol. While small amounts of glycerol were observed in the *DOG* overexpression strains in the *gpp1Δ gpp2Δ* background, no glycerol was detected in the *gpd1Δ gpd2Δ* mutant background. This indicates that overexpression of the *DOG* enzymes can rescue the osmosensitive phenotype of the *gpd1Δ gpd2Δ* mutant independent of glycerol production. We also did not observe a drop in glycerol levels in the *gpp1Δ gpp2Δ dog1Δ dog2Δ* as compared to the *gpp1Δ gpp2Δ* mutant, indicating that the Dog enzymes are not involved in glycerol biosynthesis. This indicates that Dog enzymes have a distinct substrate and their function within the cell remains undiscovered.

**IMPORTANCE** Yeast stress tolerance is an important characteristic that is studied widely, not only regarding its fundamental insights but also for its applications within the biotechnological industry. Here, we investigated the function of two phosphatase encoding genes, *DOG1* and *DOG2*, which are induced as part of the general stress response pathway, but their natural substrate in the cells remains unclear. They are known to dephosphorylate the non-natural substrate 2-deoxyglucose-6-phosphate. Here, we show that overexpression of these genes overcomes the osmosensitive phenotype of mutants that are unable to produce glycerol. However, in these overexpression strains, very little glycerol is produced indicating that the Dog enzymes do not seem to be involved in a previously predicted alternative pathway for glycerol production. Our work shows that overexpression of the DOG genes may improve osmotic and ionic stress tolerance in yeast.

**KEYWORDS** *Saccharomyces cerevisiae*, osmotic stress, ionic stress, glycerol biosynthesis, *DOG* genes

When *Saccharomyces cerevisiae* cells are grown on high concentrations of fermentable sugars, such as glucose or fructose, or under anaerobic conditions, they produce glycerol as a by-product. Biosynthesis of glycerol is important to regenerate NAD+, as this cofactor is required in the cytosol for several enzymes acting in different

Address correspondence to Patrick Van Dijck, patrick.vandijck@kuleuven.be.

Chinmayee Awasthy and Zeinab Abdelmoghis Hefny contributed equally to this article. Zeinab Abdelmoghis started the project and found the link between DOG genes and glycerol metabolism and Chinmayee finished the work and wrote the first draft of the paper.

The authors declare no conflict of interest.

See the funding table on p. 11.

metabolic pathways, including glycolysis (1–3). Furthermore, accumulation of glycerol is important as it functions as an osmoprotectant molecule, necessary for yeast cells to survive hyperosmotic or hyper-ionic stress conditions (4, 5). In the classical pathway of glycerol biosynthesis (Fig. 1), dihydroxyacetone phosphate (DHAP) is reduced into glycerol 3-phosphate (Gly-3P) by a pair of cytosolic NADH-dependent iso-enzymes encoded by the Gly-3P dehydrogenase encoding genes *GPD1* and *GPD2*. Gly-3P is further dephosphorylated into glycerol by Gly-3P phosphatase enzymes encoded by *GPP1 (RHR2)* and *GPP2 (HOR2)*.

The expression of the glycerol biosynthesis genes is regulated by the activation of the high osmolarity glycerol (HOG) response pathway. This mitogen-activated protein kinase (MAPK) pathway is essential when cells are placed under osmotic, ionic, or copper stress (6, 7). It was also shown that heat stress results in glycerol efflux which then activates the HOG pathway, depicting that different types of stress play a role in activating a common pathway (8). Yeast cells that are exposed to hyperosmotic, hyper-ionic, or heat stress achieve osmoregulation by increasing glycerol production and accumulating it inside the cells to overcome the extracellular dysbiosis in the solute to solvent ratio. However, aside from the HOG MAP kinase pathway that is crucial for glycerol production, other cellular processes, such as involvement of trehalose, and the TORC2-Ypk and the PP2A phosphatase pathways in the maintenance of cellular homeostasis upon osmotic stress conditions have been demonstrated (9).

To investigate the role of glycerol biosynthesis during specific stress responses, the Adler group, as well as some other groups, generated the *gpp1Δ gpp2Δ* double deletion strain. However, they observed that despite blocking the conversion from Gly-3P to glycerol, there were still low amounts of glycerol present in this mutant strain (5, 10). This indicated that either there are additional homologous or analogous genes that can take over the function of *GPP1* and *GPP2*, or there is another bypass route resulting in the biosynthesis of glycerol, or alternatively, recovering glycerol from lipids.

The presence of a novel alternative pathway for producing glycerol by dephosphory-lating DHAP into dihydroxyaceton (DHA), further getting reduced into glycerol, was first hypothesized by Adler et al. (11) in the salt-tolerant yeast *Debaryomyces hansenii* and by Norbeck and Blomberg (4) in *S. cerevisiae*, more than 25 years ago. Later, several reviews have also highlighted the presence of putative phosphatases that could potentially dephosphorylate DHAP into DHA, hence biosynthesizing glycerol via a novel alternative route (12–15).

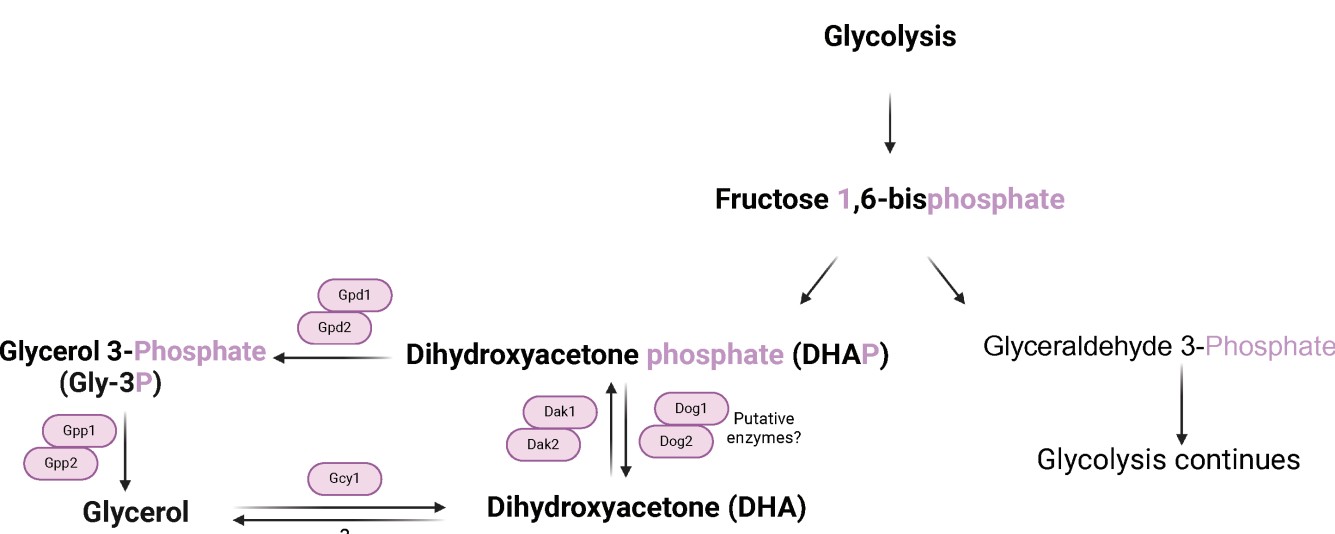

**FIG 1** Schematic representation of the classical pathway of glycerol biosynthesis. Gpd1 and Gpd2, Gly-3P dehydrogenases 1 and 2; Gpp1 and Gpp2, Gly-3P phosphatases 1 and 2; Gcy1, glycerol dehydrogenase; Dak1 and Dak2, dihydroxyacetone kinases 1 and 2; Dog1 and Dog2, 2-DG6P phosphatases 1 and 2.

The Dog enzymes are most closely related to the Gly-3P phosphatases (Gpp1 and Gpp2) with the signature catalytic motif DxDxT present at their N-termini (16, 17). They dephosphorylate multiple substrates with the highest activity toward the non-natural substrate 2-deoxyglucose-6-phosphate (2-DG6P) and hence, their specific substrate within the cell is still unknown, making their biological function an enigma (18). Dog1 and Dog2 are functionally different from each other despite having 92% sequence identity. For instance, *DOG2* has a stress-response element (STRE) motif (consensus sequence 5′-AGGGG-3′) in its promoter, which *DOG1* does not have. STRE elements are bound by Msn2 and Msn4 that are translocated to the nucleus upon environmental stress conditions, such as absence of glucose, and this binding results in the upregulation of the gene downstream to it (19). The Msn2 and Msn4 transcription factors are downstream in the cAMP-PKA pathway in *S. cerevisiae*. On the other hand, the expression of the *DOG2* gene is strongly upregulated by 2-DG6P, and this upregulation is mediated by different pathways, such as the stress response pathway mediated by the p38 MAPK ortholog Hog1, the cell wall integrity pathway mediated by the MAPK Slt2, and the unfolded protein response pathway triggered by 2-DG-induced Endoplasmic reticulum (ER) stress (20). *In vitro*-produced enzymes also showed a higher activity of Dog2 toward fructose-1-P compared to Dog1 (18). However, Kuznetsova et al. (18) report that DHAP is not an active substrate for Dog enzymes, at least *in vitro*, hence overruling the above-mentioned possibility.

As the Gpp and Dog enzymes both belong to the haloacid dehalogenase (HAD)-like group of phosphatases and as they share the DxDxT motif, we have investigated a potential role of the *DOG* genes in the biosynthesis of glycerol. In this study, we show that the overexpression of *DOG1* or *DOG2* can suppress the osmosensitive phenotype of strains lacking either both *GPP* genes or both *GPD* genes. Although these overexpression strains partially restored glycerol production in the *gpp1Δ gpp2Δ* double-mutant background, they did not result in an increase in glycerol biosynthesis in the *gpd1Δ gpd2Δ* double-mutant background. *In vitro*-purified DOG enzymes exhibited an inability to dephosphorylate both DHAP and Gly-3P, consistent with the findings reported by Kuznetsova et al. (18). Our work clearly shows that the *DOG* genes are not paralogs of the *GPP* genes but also that the real substrate of the Dog enzymes remains elusive.

## RESULTS

### Overexpression of *DOG1* or *DOG2* restores osmotolerance and ionic stress tolerance of the *gpp1Δ gpp2Δ* strain

Upon deleting the *GPP1* and *GPP2* genes, yeast cells exhibit hypersensitivity to both osmotic as well as ionic stress due to the lack of glycerol production (5). This osmosensitive phenotype of the *gpp1Δ gpp2Δ* strain was confirmed by their inability to grow under ionic stress, evidenced by their failure to grow beyond 0.4-M NaCl on solid medium (Fig. 2A) and in the presence of 0.8-M NaCl in liquid medium (Fig. 2B). A similar phenotype was also observed under osmotic stress, induced by 20% sorbitol on both solid and liquid media (Fig. 2C and D). To determine these concentrations, we tested increasing concentrations of NaCl (0.2 M, 0.4 M, 0.6 M, and 0.8 M) and sorbitol (12%, 15%, 18%, and 20%) as a source of ionic and osmotic stressors, respectively (Fig. S1A through G). Overexpression of *DOG1* or *DOG2* clearly restores the growth defect of the *gpp1Δ gpp2Δ* mutant under these conditions up to 0.4-M NaCl on solid medium and 0.8-M NaCl in liquid medium (Fig. 2A and B). Under osmotic stress conditions, the growth was restored up to 20% sorbitol on both solid and liquid media (Fig. 2C and D respectively).

### Overexpression of *DOG1* or *DOG2* restores osmotolerance and ionic stress tolerance of the *gpd1Δ gpd2Δ* strain

The previous experiment indicates that the Dog enzymes use Gly-3P as a substrate to suppress the osmosensitive phenotype of the *gpp1Δ gpp2Δ* mutant. If this would be the case, overexpression of the *DOG* genes should not be able to suppress the osmosensitive

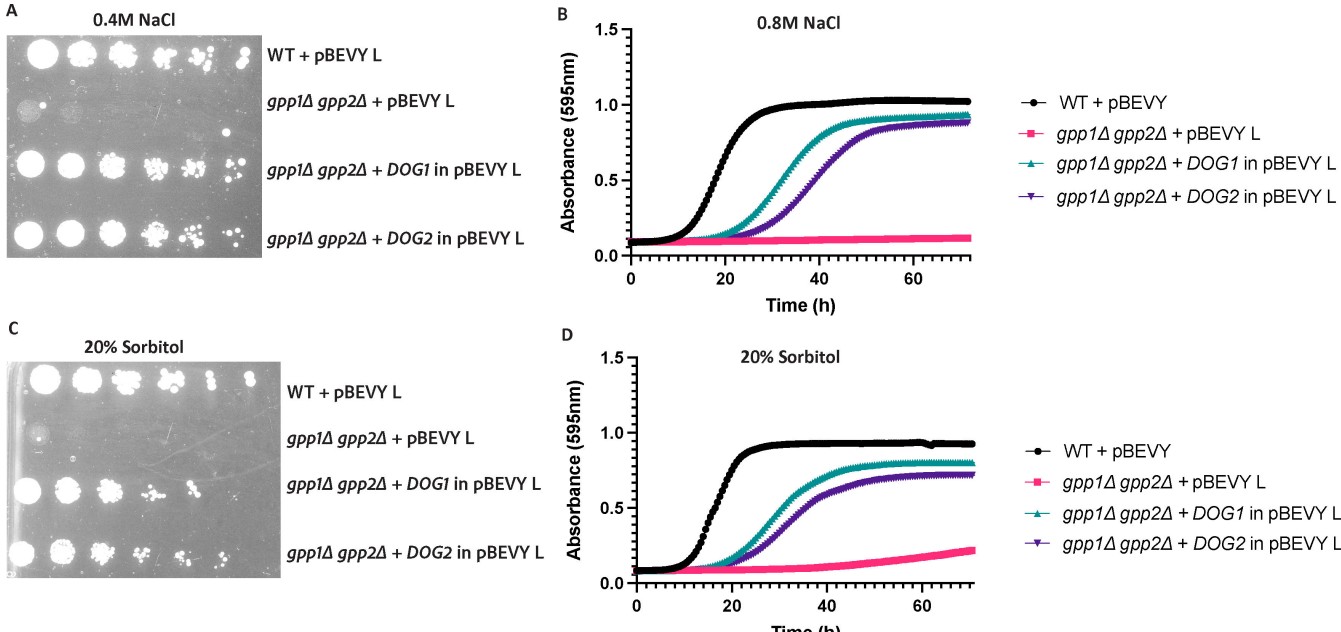

**FIG 2** Overexpression of the *DOG* genes suppress the growth-sensitive phenotype of the *gpp1Δ gpp2Δ* mutant. Growth (spot) assays was done on agar plates containing 0.4-M NaCl (panel A) and 20% sorbitol (panel C), as described in Materials and Methods. Ten-fold dilutions were spotted and visualized after 72 h of growth at 30°C. The liquid growth assays were done in medium containing 0.8-M NaCl (panel B) or 20% sorbitol (panel D), for 72 h as described in Materials and Methods. The data points are average of two independent experiments, each comprising three biological repeats. Each biological repeat is represented by the average of three technical repeats. Data are shown as average at each time point. The SEM is not shown as there was very little variation.

phenotype of the *gpd1Δ gpd2Δ* mutant, since in that strain, no Gly-3P is formed (apart from possible phospholipase activity). Figure 3 shows that the *gpd1Δ gpd2Δ* strain is sensitive to both osmotic and ionic stress, both under solid and liquid growth conditions (Fig. 3A through D). Overexpression of *DOG1* or *DOG2* in the *gpd1Δ gpd2Δ* strain showed a clear suppression of the ionic and osmotic stress-sensitive phenotype. The overexpression strains could recover the growth defective phenotype of the *gpd1Δ gpd2Δ* strain under ionic stress conditions up to 0.3-M NaCl on solid medium (Fig. 3A) and 0.2 M on liquid medium (Fig. 3B). Under osmotic stress conditions, the *DOG* overexpression strains show a clear suppression of the defective growth phenotype up to 12% sorbitol on both solid and liquid medium (Fig. 3C and D, respectively). The cells ceased to grow when the stress concentrations were increased than the values mentioned above (Fig S2A and B). These data seem to suggest that the Dog enzymes do not use Gly-3P as a substrate.

## Intracellular glycerol concentration increased in *gpp1Δ gpp2Δ* strains, but not in the *gpd1Δ gpd2Δ* strain upon overexpression of *DOG1* and *DOG2* under ionic stress conditions

Since we observed a clear suppression of the growth defect of the deletion strains (as mentioned above) upon overexpressing the *DOG* genes, we determined the intracellular glycerol levels in these strains under ionic stress conditions. Overexpression of the *DOG* genes in the *gpp1Δ gpp2Δ* strain resulted in a significant increase in intracellular glycerol levels as compared to the deletion strain expressing the empty vector, when subjected to 0.8-M NaCl (Fig. 4A and B). This could explain the suppression of the ionic stress tolerance in the overexpression strains. However, despite the evident restoration of the ionic stress tolerance in the strains overexpressing the *DOG* genes in the *gpd1Δ gpd2Δ* background, there was a surprising absence of intracellular glycerol in these strains during exposure to ionic stress induced by 0.2-M NaCl (Fig. 4C).

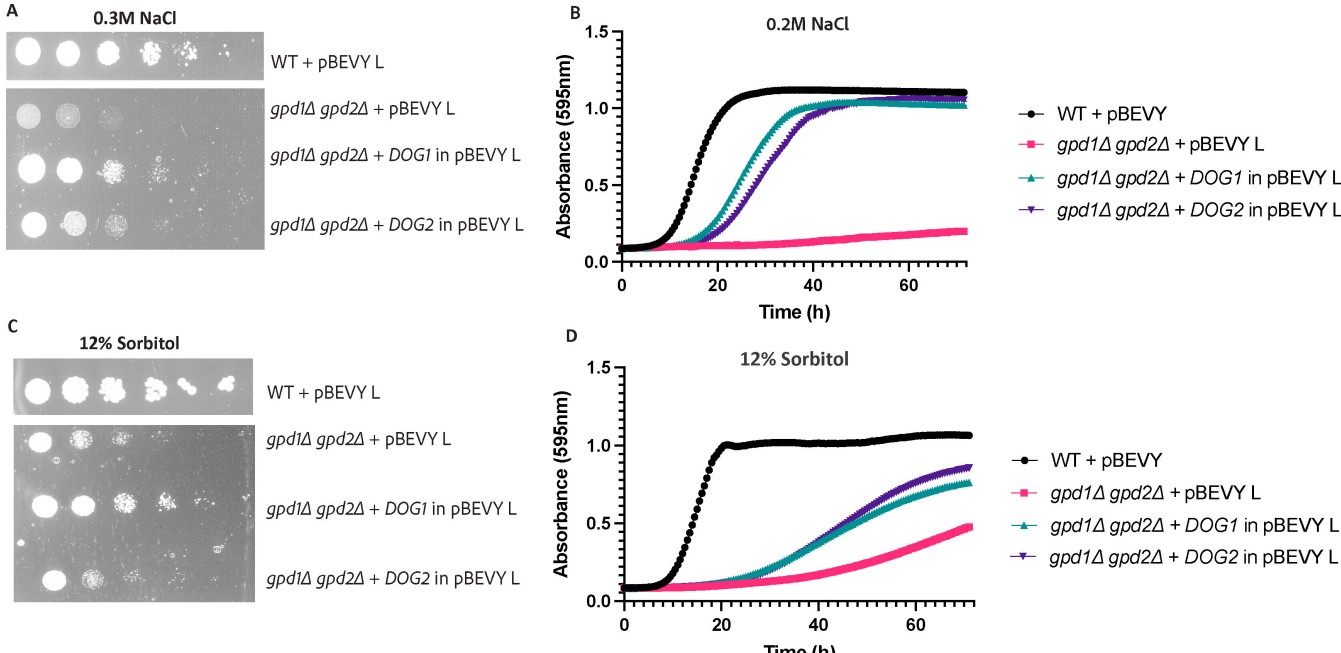

**FIG 3** Overexpression of the *DOG* genes suppress the growth-sensitive phenotype of the *gpd1Δ gpd2Δ* mutant. Growth (spot) assays was done on agar plates containing 0.3-M NaCl (panel A) and 12% sorbitol (panel C), as described in Materials and Methods. Ten-fold dilutions were spotted and visualized after 72 h of growth at 30°C. The liquid growth assays were done in medium containing 0.2-M NaCl (panel B) or 12% sorbitol (panel D), for 72 h as described in Materials and Methods. The data points are average of two independent experiments, each comprising three biological repeats. Each biological repeat is represented by the average of three technical repeats. Data are shown as average at each time point. The SEM is not shown as there was very little variation.

## *DOG1* and *DOG2* are no paralogs of *GPP1 or GPP2*

To determine whether the *DOG* genes share a similar function as Gpp1 or Gpp2, we deleted *DOG1 and DOG2* in the *gpp1Δ gpp2Δ* background to test the phenotype between this quadruple deletion strain and the *gpp1Δ gpp2Δ* strain, under both ionic and osmotic stress conditions. There was no observable difference in growth phenotype during ionic stress until 0.4-M NaCl (Fig. S3). To further validate this, intracellular glycerol levels were measured in these strains, when grown at the same NaCl concentration. Intriguingly, the glycerol concentration in this quadruple deletion strain was the same as that in the *gpp1Δ gpp2Δ* strain (Fig. 4D), that is, trace amounts of glycerol were still present, but the *DOG* genes had no observable role in its production.

## Dog proteins do not dephosphorylate DHAP

To determine the catalytic activity of the enzymes Dog1 and Dog2 in hydrolyzing the phosphate group of the substrates Gly-3P and DHAP, purified Dog1 and Dog2 proteins were used (18). The artificial sugar phosphate analog, 2-DG6P was used as a control. The assay was performed at a substrate concentration of 1 mM for the abovementioned substrates. The phosphate liberated was measured as µM of free phosphate released after the reaction. Both Dog1 and Dog2 showed high dephosphorylating activity toward 2-DG6P, as expected, with Dog1 showing higher catalytic activity toward the substrate than Dog2 (Fig. S4A and B, respectively). However, the Dog enzymes could not use DHAP or Gly-3P as a substrate (Fig. S4), confirming earlier observations (18). These data confirm that the Dog enzymes are not paralogs of the Gpp enzymes, but how they were able to suppress the osmosensitive phenotype of the glycerol biosynthesis mutants remains unclear.

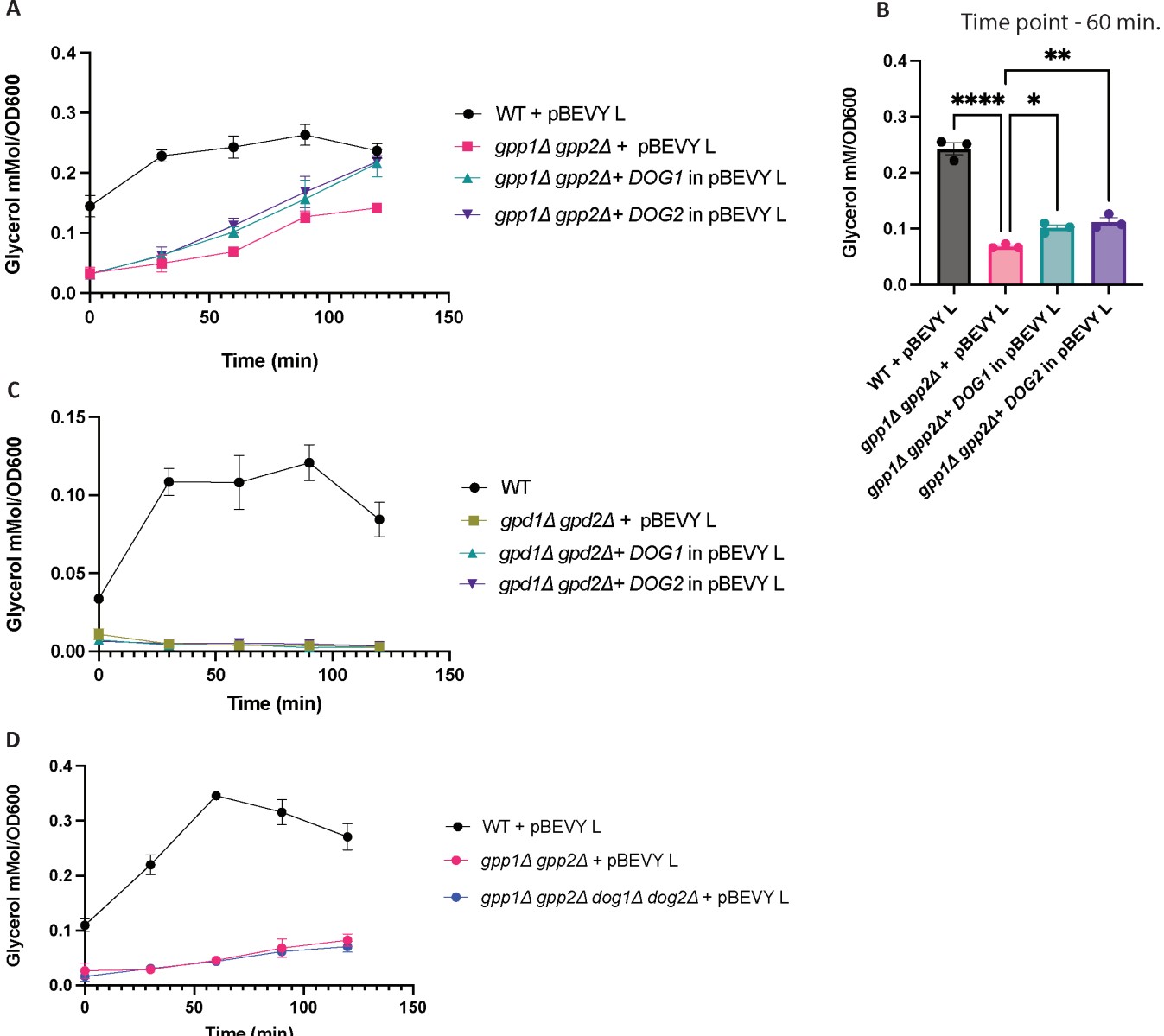

**FIG 4** Intracellular glycerol levels. The intracellular glycerol levels in (A) *gpp1Δ gpp2Δ* and *DOG* overexpressing strains in this background, (C) in *gpd1 Δ gpd2Δ* and *DOG* overexpressing strains in this background, and (D) *gpp1Δ gpp2Δ dog1Δ dog2Δ* and *DOG* overexpressing strains in this background were determined at time points 0, 30, 60, 90, and 120 minutes upon induction of stress. For *gpp1Δ gpp2Δ* and *gpp1Δ gpp2Δ dog1Δ dog2Δ* strains, 0.8 M of NaCl was added at time point zero, and for *gpd1Δ gpdΔ*, 0.2-M of NaCl was added. (B) Glycerol levels were determined in the *gpp1Δ gpp2Δ* and corresponding *DOG* overexpression strains. The assay was performed as described in Materials and Methods. Statistical analysis shown in panel B was performed by one-way ANOVA, with Bonferroni correction (comparison to *gpp1Δ gpp2Δ*). **, $P < 0.01$; ***, $P < 0.001$; ****, $P < 0.0001$. Data are shown as averages and SEM.

## DISCUSSION

While there have been some advancements in understanding 2-DG6P phosphatases (Dog1 and Dog2) since their initial characterization by Martin and Heredia in 1977 (21), the precise function and substrate of these HAD enzymes remain elusive. In this study, we explored whether the Dog enzymes could be involved in the glycerol bypass pathway catalyzing the step from DHAP to DHA and further to glycerol. However, based on our data, this does not seem to be the case. Here, we show that the overexpression of 2-DG6P phosphatases (*DOG1* and *DOG2*) restores the defective growth phenotype, under osmotic or ionic stress conditions, of both *gpp1Δ gpp2Δ* and *gpd1Δ gpd2Δ* strains. While,

in the *gpp1Δ gpp2Δ* strain, this could be explained by the small amounts of glycerol observed in the *DOG* overexpression strains, no glycerol was produced in the *gpd1Δ gpd2Δ* strain upon overexpression of either *DOG1* or *DOG2*. The glycerol present in the former strain does not seem to come from enzymatic activity of the Dog enzymes, as the quadruple *gpp1Δ gpp2Δ dog1Δ dog2Δ* strain produced the same amount of glycerol as the *gpp1Δ gpp2Δ* strain. Despite restoring the defective osmotic or ionic stress growth phenotype in *gpp1Δ gpp2Δ* or *gpd1Δ gpd2Δ* strains, the overexpression of *DOG* genes was unable to achieve the same effect in WT when subjected to ionic stress in the form of 0.8 M, 1 M, and 1,5 M of NaCl (Fig. S5A through C). This indicates that the suppression of the growth defects by *DOG* overexpression was specific to the strains where the glycerol pathway was blocked, although without any direct link to glycerol biosynthesis.

This paper confirms that the Gpp enzymes are different from the Dog enzymes regarding their enzymatic activity, which aligns with the previous study depicting that Gpp enzymes do not have 2-DG6P as a substrate (5) and that Dog enzymes do not have Gly-3P as a substrate (18). However, the native substrate for the Dog enzymes is still unclear. However, they must be involved in the production of a compatible solute, as the overexpression of these genes restored the osmotic and ionic stress sensitivity of the *gpd1Δ gpd2Δ* strain without producing any glycerol. Another important compatible solute in yeast cells is trehalose. Interestingly, the Dog enzymes share one of the phosphohydrolase motifs in trehalose-6P (T6P) phosphatase, encoded by *TPS2*, catalyzing the conversion of T6P to trehalose. Although the *tps2Δ dog1Δ dog2Δ* strain resulted in more T6P compared to the *tps2Δ* strain, overexpression of the *DOG* genes in the *tps2Δ* mutant did not restore trehalose levels (data not shown) or growth at 37°C (Fig. S6). These data are confirmed by the absence of enzymatic activity toward T6P of the purified Dog enzymes as shown by Kuznetsova et al. (18). The higher T6P levels in the triple mutant is probably because absence of the *DOG* genes may affect the amount of substrate available or may affect the general stress response pathways, further increasing the expression and/or activity of the T6P synthase enzyme, encoded by *TPS1*. This could be responsible for the higher T6P levels in the triple mutant.

A major result is that overexpression of the *DOG* genes results in some glycerol production in the *gpp1Δ gpp2Δ* strain but without using Gly-3P or DHAP as a substrate, thereby showing that the *DOG* genes do not play a role in the alternative route of glycerol biosynthesis in *S. cerevisiae*, a pathway proposed by several groups (4, 12–15). Further research will be required to elucidate the exact function of the Dog enzymes in *S. cerevisiae*. Unbiased metabolomics in wild-type, mutant, and overexpression strains could potentially provide more insight into their function. It is possible that the Dog enzymes have a more pleiotropic role, providing free phosphate to the yeast cells from any possible source. It was previously proposed that glycerol production in *S. cerevisiae* may also be important for regeneration of free inorganic phosphate used in glycolysis (22). The Dog enzymes could be involved in providing free phosphate from other sources, by liberating it from several sugar phosphates or by incorporating it from outside of the cells, for example, through the induction of *PHO84*.

## MATERIALS AND METHODS

### Strains

*S. cerevisiae* strains used in this study are listed in Table 1.

### Construction of mutants

All deletion strains were generated in the W303-1A background by replacing the gene with an antibiotic marker cassette using the plasmids mentioned in Table 2 (24). The different antibiotic resistance markers were amplified using the primers mentioned in Table 2. These primers contained tail homologs to the *GPP*, *GPD*, or *DOG* genes flanking regions. The amplified PCR fragments were directly transformed into the wild-type strain

**TABLE 1**  Strains used in this study

| Strain | Relevant genotype | Source |
|---|---|---|
| W303 1A | Mata *can1-100 leu2-3,112 trp1-1 ura3-1 ade2-1 his3-11* | (23) |
| *gpp1Δ gpp2Δ* | MATa *gpp1Δ::kan MX4 gpp2Δ::NAT* | This study |
| *gpd1Δ gpd2Δ* | MATa *gpd1Δ::Kan MX gpd2Δ::NAT* | This study |
| *dog1Δ dog2Δ* | MATa *dog1Δ::hph1 dog2Δ::Kan MX* | This study |
| *dog1Δ dog2Δ gpp1Δ gpp2Δ* | MATa *gpp1Δ::kan MX4 gpp2Δ::NAT dog1Δ dog2Δ::hph1* | This study |

using the Gietz transformation protocol (25), followed by plating on selective media consisting of Yeast extract-Peptone-Dextrose (YPD) [1% yeast extract (Merck), 2% Bacto Peptone (Oxoid), 2% glucose] and agar (2% Difco granulated agar), containing antibiotics corresponding to the antibiotic marker cassette in the transformed plasmid. The genotype of transformants was confirmed by Sanger sequencing (Eurofins Genomics). Additionally, the disruption of each gene was also confirmed via PCR by using primers in the gene, in the cassette, in and out of the cassette, and upstream and downstream of the cassette.

## Construction of *DOG1* and *DOG2* overexpression strains

To generate *DOG1* and *DOG2* overexpression strains, plasmid pBEVY-L was used (26). This plasmid possesses leucine as the auxotrophic selection marker, and the *DOG* genes were expressed under the control of the *ADH1* promoter. To accomplish this, primers fwdDOG1 5′-<u>CAAGCATACAATCAACTCCCCGGG</u>ATGGCAGAATTTTCAGCTGAT-3′ and revDOG1 5′-<u>GCTTATTTAGAAGTG TCGAATTCT</u>CAGTGGTGATGGTGATGATGGGCCCTTGTCA AAGGGTTGTT-3′ were used to amplify *DOG1*. Similarly, primers fwdDOG2 5′-<u>AAGCATACA ATCAACTCCCCGG</u>

<u>G</u>ATGCCACAATTTTCAGTAG-3′ and revDOG2 5′<u>CCGCTTATTTAGAAGTGTCGA</u>GAACATC GTAT

GGGTAATCTCTCGTCAAAGGGTTGT-3′ were used to amplify *DOG2*. The tails homologous to the plasmid backbone are underlined in the primers above. The gene amplification was carried out using Q5 High-Fidelity DNA Polymerase (New England Biolabs), using genomic DNA from the wild-type (W303-1A) strain as the template for the PCR reaction. This DNA was extracted using phenol/chloroform/isoamyl alcohol (PCI) followed by ethanol precipitation. The plasmid was cut using the restriction enzymes SmaI and EcoRI, and the genes *DOG1* or *DOG2* were cloned in between them. The PCR fragment, containing the overhanging tails, was ligated with the digested plasmid using the NEBuilder HiFi DNA Assembly Master Mix which was then transformed into *Escherichia coli* TOP10 (NEB) using the heat-shock method. After recovery, the cells were grown in Lysogeny Broth (LB) medium, containing 100-µg/mL ampicillin at 37°C. The plasmids were then isolated from single colonies using the NucleoSpin Plasmid EasyPure Kit. The insertion of both genes was confirmed via Sanger sequencing (Eurofins Genomics), and the correct plasmid was further transformed into WT (W303 1A), *gpp1Δ gpp2Δ*, *gpd1Δ gpd2Δ*, *dog1Δ dog2Δ*, and *gpp1Δ gpp2Δ dog1Δ dog2Δ* strains using the Gietz transformation protocol (25). As controls, the empty vector was transformed into these strains. The transformants were selected on (Complete Synthetic Medium) CSM-leucine agar plates supplemented with 100-mg/L adenine and 2% glucose.

## Growth conditions

The deletion strains were grown in YPD [1% yeast extract (Merck), 2% Bacto Peptone (Oxoid), 2% glucose] medium. For solid medium, 2% Difco granulated agar was added. All overexpression strains were cultured in CSM-leucine (MP Biomedicals) supplemented with 100-mg/L L-adenine (Sigma-Aldrich). Glucose was added to a final concentration of 2%. The pH was adjusted to 5.5 for liquid media and 6.5 for media containing agar. The deletion strains were stored at −80°C in stock media composed of YP and 30% glycerol.

**TABLE 2** Primers used to construct deletion strains for this study

| Deletion strain | Plasmid used | Primers used |
|---|---|---|
| gpp1Δ | - | From lab collection |
| gpp2Δ | pTOPO-A1-G1-B-NAT-P-G1-A2(p72) | Forward-5′-AATAGCGCCAACCAGCTAGTGTTTACCAGATCAGTGGAAAAACATAAAACAATAAAAACAATATTCGGAGTGGTCGGCTGGAGATCGG-3′ <br> Reverse-3′-GTTTTGTGGCGAATATAGAATAGGACTGTATCTGAGAATTATTACTCAAATATGTTCGATTTTAGAGGAAGCCGTTATGGCGGGCATC-5′ |
| gpd1Δ | pTOPO-A1-G1-B-KanMX-P-G1-A2(p71) | Forward-5′-AGACATCAAGAAACAATTGTATATTGTACACCCCCCCCCTCCACAAACACAAATATTGATAATATAAAGGTGGTCGGCTGGAGATCGG-3′ <br> Reverse-3′-TGAATATGATATAGAAGAGCCTCGAAAAAAGTGGGGGAAAGTATGATATGTTATCTTTCTCCAATAAATAGCCGTTATGGCGGGCATC-5′ |
| gpd2Δ | pTOPO-A1-G1-B-NAT-P-G1-A2(p72) | Forward-5′-TAAGTTTATGTATTTTGGTAGATTCAATTCTCTTTCCCTTTCCTTTTCCTTCGCTCCCCTTCCTTATCAGTGGTCGGCTGGAGATCGG-3′ <br> Reverse-3′-TAATGATAAATTGGTTGGGGGAAAAAGAGGCAACAGGAAAGATCAGAGGGGGAGGGGGGGGGAGAGTGTAGCCGTTATGGCGGGCATC-5′ |
| dog1Δ | pJET1,2-B-hph-p | Forward-5′-AACGTTCTGATGGAGATTGTTGGTTACGTTGCCACTCACGTAAGAAGTTC GTGGTCGGCTGGAGATCGG-3′ <br> Reverse-3′-GCATTCCAACGAAATACAGTTGATGGAAGTGTTTTGCGCTAAGGAAGATTAGCCGTTATGGCGGGCATC-5′ |
| dog2Δ | pJET1,2-B-kanMX4-P | Forward-5′-ATAGTATTAGCCAACACGTTATCGATACATTTACTGCTATATACATAAAAAATTTACGGTGGTCGGCTGGAGATCGG-3′ <br> Reverse-3′-GTAATCAGCCCACTGATCAAGCCTTCGGCGCGGTTGTTCAACCACACGATCTGTATCAAAGAGCCGTTATGGCGGGCATC-5′ |
| dog1Δ dog2Δ in gpp1Δ gpp2Δ | pTOPO-A1-G2-B-HPH-P-G2-A2(p79) | Forward-5′-ATTTCATACTCACTATAAGAAATCATACGCAGTTCAACTTTTGCTTTTACATACAATTTTATCTATATAAGCCGTTATGGCGGGCATC-3′ <br> Reverse-3′-AATGATCGCGATTATTCTGTTGGAAATAACGTTCTGATGGAGATTGTTGGTTACGTTGCCACTCACGTAGTGGTCGGCTGGAGATCGG-5′ |

## Phenotype assays

All strains were grown in CSM-leucine medium supplemented with 100-mg/L adenine and 2% glucose. The pre-cultures were grown for 38–40 h, that is, until the cells reached stationary phase at 30℃. These cultures were washed thrice in phosphate-buffered saline (PBS), and Optical Density (OD) was measured at 600 nm. For the liquid growth assay, these cultures were then diluted to a starting $OD_{600}$ of 0.05 in liquid medium with NaCl or sorbitol as ionic and osmotic stressors, respectively, and *Multiscan FC* from Thermo Scientific was further used to carry out this experiment in a 96-well microtiter plate (Greiner). For spot assays, the cultures were diluted to an $OD_{600}$ of 1 in PBS, and 5 µL from this was spotted on agar plates with media containing the abovementioned stressors. The growth temperature was 30℃ under all conditions. The data were plotted as average and standard error of mean (SEM) using GraphPad Prism version 10.0.3.

## Intracellular glycerol measurements

Intracellular glycerol concentrations were determined as mentioned in Dunayevich et al. (8) with modifications. Cells were grown for 38–40 h (pre cultures) in CSM-leucine medium supplemented with 100-mg/L adenine and 2% glucose. They were then diluted to an OD of 0.2 in fresh medium and grown to mid-log phase. At this stage, NaCl according to the strain's threshold for stress was added to these cultures. For *gpp1Δ gpp2Δ* and *gpp1Δ gpp2Δ dog1Δ dog2Δ*, 0.8 M of NaCl was added as the stressor, and for *gpd1Δ gpdΔ*, 0.2 M of NaCl was added as the stressor. One-milliliter aliquots were collected at timepoints 0, 30, 60, 90, and 120 minutes. Immediately after this, cells were harvested by centrifugation for 1 minute at 4,000 g, resuspended in 1 mL of boiling water and incubated at 100℃ for 10 minutes. Samples were cooled down on ice for 10 minutes and subsequently centrifugated at 14,000 g. Supernatants were used to measure glycerol and were stored at −20℃. $OD_{600}$ was determined at all time points. Glycerol concentration was determined using a commercial kit following manufacturer's instructions (MAK117, Sigma-Aldrich). The amount of glycerol was normalized to $OD_{600}$. The data were plotted as average and SEM using GraphPad Prism version 10.0.3.

## Protein expression and extraction

The constructs of Dog1 and Dog2 cloned in a bacterial expression vector were kindly provided by Dr. Yakunin (Bangor University, UK and U. Toronto, Canada) (18). These were then transformed into *E. coli* BL21(DE3) competent cells, selected on LB agar media supplemented with 100-mg/mL ampicillin and were incubated overnight at 37°C. The colonies were then transferred and grown in 1-L fresh medium at 37°C until an $OD_{600}$ of 0.5–0.7. These cells were induced with Isopropyl β- d-1-thiogalactopyranoside (IPTG) at a final concentration of 0.5 mM and were incubated overnight at 16°C, shaking at 150 rpm. The next day, cells were collected by centrifugation for 45 minutes at 4,600 rpm at 4°C. The extracts were kept on ice from then on. To lyse the cells, a lysis buffer composed of 20-mM Tris-HCL (pH 8.0), 100-mM NaCl, 10% glycerol, and 20-mM imidazole was used, and Hen eggwhite lysozyme (20 mg/mL) along with 100-mM pefabloc was added to them. Next, the lysed cells were treated with DNaseI, sonicated (10 cycles of 30 s: 1 s pulse, 1 s rest, 40% amplitude) and centrifuged at 26,000 rpm for 25 minutes at 4°C. The lysate was filtered using a 0.45-μm filter, and the protein was purified manually.

## Protein purification

To purify the protein, a Ni-NTA column was used. This was first washed with MQ $H_2O$ and then equilibrated with the binding buffer (20-mM Tris-HCL (pH 8.0), 100-mM NaCl, 10% glycerol, and 20-mM imidazole). The lysate was then added onto it, left to rest for a couple of minutes, and the flow-through was collected from this step onward. Elution buffer containing 50-mM, 100-mM, 200-mM, 300-mM, and 500-mM imidazole was applied consecutively, and subsequently, the flow-through containing our protein of interest was collected. Based on the SDS-PAGE analysis of the presence of the desired protein, the fractions were transferred to a Macrosep Advance centrifugal device (3-kDa cut-off) and centrifuged at 4,600 rpm at 4°C for 25 minutes. Storage buffer [20-mM Tris-HCL (pH 8.0), 100-mM NaCl, 10% glycerol] was added, and the samples were centrifuged at 4,600 rpm at 4°C for 25 minutes. This step was repeated twice, and the samples were transferred to Microsep Advance centrifugal device (3-kDa cut-off) and centrifuged again at 4,600 rpm at 4°C. The final concentration of the purified protein was determined by the formula $A_\lambda = \varepsilon C$, where $A_\lambda$ is the absorbance of the protein measured via nanodrop and $\varepsilon$ is the average of the two extinction coefficients of the protein. Furthermore, this value (*C*) was multiplied by the weight of the protein in kDa giving us the final concentration of our protein of interest.

## Visualization of the purified protein

The protein samples were boiled for 5 minutes and were separated by SDS-PAGE (NuPAGE 4%–12% Bis-Tris gel; Invitrogen), using 1× NuPAGE MOPS SDS Running Buffer (Invitrogen), at a constant 180 V for 60 minutes. To quantify the protein of interest on the gel, it was stained with GelCode Blue Stain Reagent (Thermo Scientific). The protein could be visualized based on the thickness of the band at the correct size, based on the molecular weight marker proteins.

## Phosphatase assay

In this assay, the concentration of the liberated inorganic phosphate was determined using the Phosphate Assay Kit (Sigma-Aldrich) according to the manufacturer's protocol. The reaction was carried out in a 96-well microtiter plate (Greiner) with 0.01-mg/mL protein, 1 mM of the substrate, and was supplemented with 5-mM $MgCl_2$ (pH 7.5) as the cofactor. This mix was incubated for 30 minutes at room temperature, and malachite green reagent, provided in the kit, was added to stop the reaction. The samples were immediately analyzed using the Synergy H1 hybrid multimode microplate reader (BioTek). All sugar phosphates used in this experiment were stored at −20°C. 2-DG6P sodium salt, DHAP dilithium salt, and sn-Gly-3P lithium salt were purchased from

Sigma-Aldrich. The assay was carried out in triplicates. The data were plotted as average and SEM using GraphPad Prism version 10.0.3.

## Statistics

Statistical analysis was performed by one-way analysis of variance (ANOVA), with Bonferroni correction **, $P < 0.01$; ***, $P < 0.001$; ****, $P < 0.0001$ using GraphPad Prism Version 10.0.3.

## ACKNOWLEDGMENTS

This work was supported by the Fund for Scientific Research Flanders (FWO grant # G0B1820N). Z.A.H. was supported from a grant of the Schlumberger foundation (FFTF), by the Egyptian Cultural Affairs and Missions Sector and by a KU Leuven junior mobility grant (JUMO). The funders had no role in study design, data collection and interpretation, or the decision to submit the work for publication. Next, we would like to thank Dr. Paul Vandecruys and Ilse Palmans for their valuable assistance in constructing deletion and overexpression strains and Prof. Joleen Masschelein and Dr. Sofie Dekimpe for help with the protein purification work. We extend our appreciation to Prof. Alexander F. Yakunin (Bangor University, UK and U. Toronto, Canada) for kindly providing the Dog1 and Dog2 expression constructs in bacterial plasmid vectors.

The Egyptian Cultural Affairs and Missions Sector, The Schlumberger Foundation and the KU Leuven JUMO grant.

## AUTHOR AFFILIATIONS

[1]Laboratory of Molecular Cell Biology, Institute of Botany and Microbiology, Leuven, Belgium
[2]Biomedical MRI/MoSAIC Lab, Department of Imaging and Pathology, KU Leuven, Leuven, Belgium

## AUTHOR ORCIDs

Chinmayee Awasthy  http://orcid.org/0000-0002-6148-0701
Wouter Van Genechten  http://orcid.org/0000-0002-8562-8029
Uwe Himmelreich  http://orcid.org/0000-0002-2060-8895
Patrick Van Dijck  http://orcid.org/0000-0002-1542-897X

## FUNDING

| Funder | Grant(s) | Author(s) |
| --- | --- | --- |
| Fonds Wetenschappelijk Onderzoek (FWO) | G0B1820N | Patrick Van Dijck |

## ADDITIONAL FILES

The following material is available online.

### Supplemental Material

**Supplemental material (Spectrum00136-24-s0001.pdf).** Fig. S1 to S6.

### Open Peer Review

**PEER REVIEW HISTORY (review-history.pdf).** An accounting of the reviewer comments and feedback.

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
