## [Reviewer comments · Microbiology Spectrum]

Microbiology Spectrum

Involvement of 2-deoxyglucose-6-phosphate phosphatases in facilitating resilience against ionic and osmotic stress in *Saccharomyces cerevisiae*

Chinmayee Awasthy, Zeinab Hefny, Wouter Van Genechten, Uwe Himmelreich, and Patrick Van Dijck

Corresponding Author(s): Patrick Van Dijck, Associatie KU Leuven

Review Timeline:

Submission Date:	March 23, 2024
Editorial Decision:	May 24, 2024
Revision Received:	June 1, 2024
Accepted:	June 16, 2024

Editor: Erik Hom

Reviewer(s): The reviewers have opted to remain anonymous.

Transaction Report:

DOI: <https://doi.org/10.1128/spectrum.00136-24>

Re: Spectrum00136-24 (Involvement of 2-deoxyglucose-6-phosphatohosphatases in facilitating resilience against ionic and osmotic stress in *Saccharomyces cerevisiae*)

Dear Prof. Patrick Van Dijck:

Thank you for the privilege of reviewing your work. Below you will find my comments, instructions from the Spectrum editorial office, and the reviewer comments.

Please carefully proof-read your submission and iron out the language issues as suggested by the reviewer. In your cover letter to me, please confirm what you have revised.

Revision Guidelines

Sincerely,
Erik Hom
Editor
Microbiology Spectrum

The manuscript by Awasthy et al., discusses the role of DOG genes in *S. cerevisiae* vis a vis their putative involvement in glycerol biosynthesis. The manuscript in its current form consists of preliminary data about DOG genes, and does not bring forward any additional information other than what is known on these genes. Additionally, concluding that DOG genes may have a robust industrial impact seems overstretched based on the current data. As discussed by the authors, performing genome wide profiling experiments on the various mutant strains is something that should be done that may be able to provide more novel information on the role of the DOG genes. Moreover, the manuscript needs to be revamped totally for its flow and language. As it stands, this study needs to be strengthened in terms of data. In terms of language used and the flow there are many flaws; a few have been listed below. The authors also should have a look at the following concerns:

1. The title should be

DOG genes facilitate resilience against ionic and osmotic stress in *Saccharomyces cerevisiae*
OR

Involvement of *DOG* genes in facilitating resilience against ionic and osmotic stress in *Saccharomyces cerevisiae*

In any case, the term **yeast** should be omitted from the title. **Moreover, the authors should not use the abbreviated nomenclature for DOG genes at least in the title.**

2. The authors should show a schematic of the glycerol biosynthesis pathways for the reader.

3. **Line 23-25:** “..... superfamily encoding enzymes that dephosphorylate multiple.....”

4. **Line 27-28:** should be rephrased.

5. **Line 71-72:**play a role in activating a common pathway”

6. **Line 117:** should be “DOG”

7. **Line 199 :** ..”not..”

8. **Lines 131-133:** should be rephrased

9. **Fig. 2C:** Any reason why DOG2 does not rescue the sensitivity of the mutant strains on solid medium, while it does so in liquid medium.

10. Fig.3: While describing the increase in the glycerol content, the authors should mention the fold-increase? The fold-increase does not look significant in the figure. Moreover, this comparison should have included the wild type strain overexpressing the DOG genes to see if this fold increase is specific to the mutant overexpressing the genes? What if the DOG genes when overexpressed in the wild type also result in an increase in the glycerol content, suggestive of them operating in a totally independent pathway?! It would also be interesting also to check the growth of the WT overexpressing DOG on these stressors.

11. Figures 3A and D can be merged, and therefore the two sections of the result should be merged too.

12. The authors should combine the results with discussions instead of keeping them separate.

The manuscript by Awasthy et al., discusses the role of DOG genes in *S. cerevisiae* vis a vis their putative involvement in glycerol biosynthesis. In its revised for, the authors have addressed the comments satisfactorily, albeit the syntactical errors. Some of the statements again need to be rephrased, for which the authors should give the manuscript a thorough read one more time. The following are few such examples:

Abstract: Should be rephrased as “These enzymes of the Haloacid dehalogenase (HAD) superfamily are known to utilize the non-natural 2- deoxyglucose-6-phosphate as their substrate. However, their physiological substrate, and hence their biological role remains elusive. ”

Figure 1: The references within the figure should be removed, and a legend should be provided with the full names of the enzymes mentioned.

Lines 75-79: Rephrased to “However, aside from the HOG MAP kinase pathway that is crucial for glycerol production, other cellular processes, such as involvement of trehalose, and the TORC2-Ypk and the PP2A phosphatase pathways in the maintenance of cellular homeostasis upon osmotic stress conditions has been demonstrated (9). ”

Reply to the referee comments

The manuscript by Awasthy et al., discusses the role of DOG genes in *S. cerevisiae* vis a vis their putative involvement in glycerol biosynthesis. In its revised for, the authors have addressed the comments satisfactorily, albeit the syntactical errors. Some of the statements again need to be rephrased, for which the authors should give the manuscript a thorough read one more time. The following are few such examples:

Abstract: Should be rephrased as “These enzymes of the Haloacid dehalogenase (HAD) superfamily are known to utilize the non-natural 2- deoxyglucose-6-phosphate as their substrate. However, their physiological substrate, and hence their biological role remains elusive.”

The sentence has been rephrased.

Figure 1: The references within the figure should be removed, and a legend should be provided with the full names of the enzymes mentioned.

The references have been removed and the full names of the enzymes have been mentioned as figure legend.

Lines 75-79: Rephrased to “However, aside from the HOG MAP kinase pathway that is crucial for glycerol production, other cellular processes, such as involvement of trehalose, and the TORC2-Ypk and the PP2A phosphatase pathways in the maintenance of cellular homeostasis upon osmotic stress conditions have been demonstrated (9).”

The sentence, now lines 74 – 77, has been rephrased.

Apart from adapting the above-mentioned comments, we have also added Gly-3P in brackets, the abbreviation for Glycerol–3 phosphate, in Figure 1.

We also read carefully through the manuscript and adapted a few sentences.

We now also refer to supplementary figures S5 and S6 as that was not the case in the previous version.

Re: Spectrum00136-24R1 (Involvement of 2-deoxyglucose-6-phosphatase in facilitating resilience against ionic and osmotic stress in *Saccharomyces cerevisiae*)

Dear Prof. Patrick Van Dijck:

Your manuscript has been accepted, and I am forwarding it to the ASM production staff for publication. Your paper will first be checked to make sure all elements meet the technical requirements. ASM staff will contact you if anything needs to be revised before copyediting and production can begin. Otherwise, you will be notified when your proofs are ready to be viewed.

Sincerely,
Erik Hom
Editor
Microbiology Spectrum